The role of endoplasmic reticulum stress in type 2 diabetes mellitus mechanisms and impact on islet function

He Zhaxicao 1
Liu Qian 1
Wang Yan 1
Zhao Bing 1
Zhang Lumei 1
Yang Xia 2
Wang Zhigang tsszy@163.com 1 2
1 Gansu University of Chinese Medicine , Lanzhou , China
2 Tianshui Hospital of Traditional Chinese Medicine , Tianshui , China
Gould Gwyn
Electronic publication date: 2025 Mar 28
Publication date: 2025
Volume: 13
Electronic Location ID: e19192
Received 2024 Dec 18; Accepted 2025 Feb 26
Copyright: ©2025 He et al.
Copyright year: 2025
Copyright holder: He et al.
License: This is an open access article distributed under the terms of the Creative Commons Attribution License, which permits unrestricted use, distribution, reproduction and adaptation in any medium and for any purpose provided that it is properly attributed. For attribution, the original author(s), title, publication source (PeerJ) and either DOI or URL of the article must be cited.
License URL: https://creativecommons.org/licenses/by/4.0/

Keywords: Diabetes, Endoplasmic reticulum stress, Pancreatic cell, Unfolded protein response, Therapeutic strategies, Islet cell

Funding: Natural Science Foundation of Gansu Province 25JRRE013 Subject of Gansu Provincial Administration of Chinese Medicine GZKZ-2024-39 This work was supported by the Natural Science Foundation of Gansu Province (No. 25JRRE013) and the Subject of Gansu Provincial Administration of Chinese Medicine (No. GZKZ-2024-39). The funders had no role in study design, data collection and analysis, decision to publish, or preparation of the manuscript.

==============================
Type 2 diabetes mellitus (T2DM) is a globally prevalent metabolic disorder characterized by insulin resistance and dysfunction of islet cells. Endoplasmic reticulum (ER) stress plays a crucial role in the pathogenesis and progression of T2DM, especially in the function and survival of β-cells. β-cells are particularly sensitive to ER stress because they require substantial insulin synthesis and secretion energy. In the early stages of T2DM, the increased demand for insulin exacerbates β-cell ER stress. Although the unfolded protein response (UPR) can temporarily alleviate this stress, prolonged or excessive stress leads to pancreatic cell dysfunction and apoptosis, resulting in insufficient insulin secretion. This review explores the mechanisms of ER stress in T2DM, particularly its impact on islet cells. We discuss how ER stress activates UPR signaling pathways to regulate protein folding and degradation, but when stress becomes excessive, these pathways may contribute to β-cell death. A deeper understanding of how ER stress impacts islet cells could lead to the development of novel T2DM treatment strategies aimed at improving islet function and slowing disease progression.

Introduction

Diabetes has become a global public health crisis, particularly type 2 diabetes mellitus (T2DM), which is characterized by insulin resistance and dysfunction of pancreatic β-cells (Stumvoll, Goldstein & Van Haeften, 2005). The pancreas is a vital organ with both endocrine and exocrine functions. Its endocrine component is primarily composed of islet cells, including α-cells, β-cells, δ-cells, PP-cells, and ɛ-cells (Campbell & Newgard, 2021). β-cells account for 60–80% of the total islet cell population and are responsible for synthesizing and secreting insulin, a critical hormone for maintaining glucose homeostasis and regulating carbohydrate and lipid metabolism (Campbell & Newgard, 2021). Dysfunction of β-cells leads to insufficient insulin secretion, resulting in hyperglycemia and metabolic disturbances (Roder et al., 2016).

The endoplasmic reticulum (ER) is a crucial cellular organelle involved in protein synthesis, folding, glycosylation, and calcium ion storage, among other essential functions (Almanza et al., 2019; Celik et al., 2023). It provides a specialized folding environment, distinct from the cytoplasm, with a redox balance and numerous enzymes that assist in protein folding (Acosta-Alvear et al., 2024). It is also the primary site for the formation of disulfide bonds in proteins (Cooper et al., 2017). Furthermore, the ER serves as the main entry point for the secretory pathway, where nearly all secretory proteins fold (Plummer et al., 2016). ER stress is triggered when the ER’s protein-folding capacity is exceeded and misfolded or unfolded proteins accumulate (Jiang et al., 2024). To counteract this, cells activate the unfolded protein response (UPR), which decreases the production of new protein, enhances the protein-folding capacity, and promotes the degradation of misfolded proteins, thereby restoring ER homeostasis (Bernales, Papa & Walter, 2006; Hetz, Zhang & Kaufman, 2020; Oakes & Papa, 2015). However, prolonged or excessive ER stress can lead to cellular dysfunction and even trigger apoptosis (Ajoolabady et al., 2023).

Due to the high demand for insulin synthesis and secretion, the ER of β-cells is often under significant stress (Yong et al., 2021a). Since under conditions of overnutrition, obesity, or genetic predisposition leading to insulin resistance, β-cells are required to produce and secrete more insulin to maintain blood glucose balance, insulin folding in the ER involves the formation of disulfide bonds between cysteine residues in the A- and B-chains, a critical process for its correct structure. However, the accumulation of misfolded insulin, can exacerbate ER stress and trigger the UPR (Ajoolabady et al., 2023; Blanc et al., 2024; Yang et al., 2022). Persistent ER stress activates stress response pathways, such as the C/EBP homologous protein (CHOP) and caspase-12, ultimately leading to β-cell apoptosis (Fonseca, Gromada & Urano, 2011). In both type 1 and type 2 diabetes, ER stress is considered a key factor contributing to β-cell dysfunction and apoptosis. Features of diabetes, such as hyperglycemia and hyperlipidemia, aggravate ER stress, creating a vicious cycle that further impairs β-cell function (Mellado-Gil et al., 2016; Vig et al., 2021; Yong et al., 2021a). Therefore, this article systematically introduces how different pancreatic islet cells respond to ER stress during the process of diabetes, with a particular focus on how ER stress impacts β-cell function and how diabetic conditions exacerbate ER stress, which is crucial for elucidating the pathogenesis of diabetes, thereby providing new insights for developing targeted therapeutic strategies.

Audience

This review is intended for researchers whose interest are endoplasmic reticulum stress and type 2 diabetes mellitus.

Survey methodology

PubMed database was used for related literature search using the keyword “Endoplasmic Reticulum Stress” “Diabetes” “Islet Cell” “Unfolded Protein Response” and “Therapeutic Strategies”.

Islet Cells and ER Stress

The islets of Langerhans are small clusters of endocrine cells scattered within the exocrine tissue of the pancreas. These islet cells work in concert to regulate blood glucose and maintain glucose homeostasis. As shown in Fig. 1, the major cell types within the islets include β-cells, α-cells, δ-cells, PP-cells, and ɛ-cells, which play complementary and synergistic roles in blood glucose regulation through the secretion of specific hormones (Campbell & Newgard, 2021). β-cells are the predominant cell type in the islets, accounting for 60–80% of the total islet cell population and are mainly localized in the central regions of the islets (Campbell & Newgard, 2021). The primary function of β-cells is the synthesis and secretion of insulin, a key hormone involved in regulating blood glucose levels (Henquin, 2011; Szollosi et al., 2007). Insulin lowers blood glucose by promoting glucose uptake by cells, inhibiting hepatic glycogen breakdown, and suppressing gluconeogenesis (Beltowski, Wojcicka & Jamroz-Wisniewska, 2018; Petersen et al., 1998). As a complex protein, insulin synthesis involves multiple folding and modification steps, necessitating a highly developed rough endoplasmic reticulum (RER) in β-cells to support efficient protein synthesis and processing (Lee et al., 2015; Rabhi et al., 2014). This extensive RER development makes β-cells highly dependent on ER function, rendering them particularly susceptible to damage under conditions of ER stress (Sharma et al., 2015). Chronic ER stress can lead to β-cell dysfunction or apoptosis, contributing to insufficient insulin secretion, a major pathological mechanism in diseases like T2DM. α-cells account for approximately 15–20% of the islet cell population and are typically located in the peripheral regions of the islets (Campbell & Newgard, 2021). The primary function of α-cells is the secretion of glucagon, a hormone that stimulates glycogen breakdown in the liver and glucose release, thereby raising blood glucose levels during hypoglycemia. Compared to β-cells, α-cells exhibit a more prominent endoplasmic reticulum, with the length of the ER cisternae in α-cells extending 1.4 ± 0.6 µm in two-dimensional sections, compared to 0.8 ± 0.5 µm in β-cells. This suggests that the ER in α-cells is more developed and widespread than in β-cells (Like & Chick, 1970; Pfeifer et al., 2015). However, the Golgi apparatus cisternae are more pronounced in β-cells than in α-cells (Pfeifer et al., 2015). δ-cells, which constitute 5–10% of the islet cell population, are primarily located at the islet periphery (Rorsman & Huising, 2018). These cells secrete somatostatin, which regulates the secretion of both insulin and glucagon. Additionally, there are PP-cells and ɛ-cells in the islets, but their proportions are much smaller, and their primary role is in auxiliary regulation (Khan et al., 2016). PP-cells secrete pancreatic polypeptide, which regulates the exocrine function of the pancreas and gastrointestinal motility, while ɛ-cells secrete ghrelin, a hormone involved in regulating appetite (Dezaki & Yada, 2022; Khan et al., 2016). Although reports on the ER structure in these smaller populations of cells are limited, as endocrine cells in the islets, their secretion, like that of α-cells, is often calcium-dependent (Rorsman & Huising, 2018; Vierra et al., 2018), and the calcium content in the ER is a key factor influencing their secretory activity (Adriaenssens et al., 2016).

Figure 1 Each pancreatic islet consists of various types of endocrine cells.

The main cell types in the islets include β-cells, α-cells, δ-cells, and ɛ-cells, each occupying different proportions within the islet. Among them, β-cells are the predominant cell type, comprising 60% to 80% of the total islet cell population and are mainly located in the central region of the islet. α-cells account for approximately 15% to 20% of the islet cell population and are typically situated at the periphery of the islet. δ-cells make up a smaller proportion, about 5% to 10%, and are generally located at the islet’s edges. ɛ-cells are extremely scarce, representing less than 1% of the total islet cells.

The differences in the ER structure and abundance between islet cell types reflect their varying functional and protein synthesis demands. β-cells, due to their role in synthesizing and secreting large amounts of insulin, possess an exceptionally developed ER and are more sensitive to ER stress. The high level of protein synthesis in β-cells makes them more prone to the accumulation of unfolded or misfolded proteins, which triggers ER stress and the UPR. Persistent ER stress can lead to β-cell dysfunction and apoptosis, playing a critical role in the onset and progression of diabetes. In contrast, α-cells and other islet cell types, due to their lower reliance on ER function during protein synthesis, may exhibit higher tolerance to ER stress.

Mechanisms of ER Stress

The ER is a critical cellular organelle responsible for protein synthesis, folding, modification, and transport, playing a pivotal role in maintaining cellular protein homeostasis and function. Under normal physiological conditions, newly synthesized proteins fold and assemble correctly within the ER lumen. However, when the ER’s protein-folding capacity is challenged—due to factors such as calcium ion imbalance, oxidative stress, nutrient deprivation, or excessive protein synthesis—misfolded or unfolded proteins accumulate in the ER, leading to the onset of ER stress (Scheuner & Kaufman, 2008).

In pancreatic β-cells, ER stress is closely linked to islet function, especially in the context of the onset and progression of diabetes. β-cells require substantial ER function to synthesize and fold insulin. Excessive protein synthesis burden, calcium ion imbalance, and oxidative stress can exacerbate ER stress, impairing proper protein folding and causing β-cell dysfunction (Hasnain, Prins & McGuckin, 2016; Lytrivi et al., 2020). Chronic ER stress is commonly observed in diabetes, and it accelerates the loss of β-cells by promoting apoptosis and inhibiting their proliferation (Hasnain et al., 2014). Studies have shown that in diabetic models, the excessive activation of ER stress and UPR can lead to β-cell exhaustion and loss of islet function.

To cope with ER stress, cells activate a highly conserved signaling pathway known as the UPR. The primary function of the UPR is to restore ER homeostasis, ensuring cell survival. Since most proteins are synthesized on ribosomes in the cytoplasm and are then directed to the endoplasmic reticulum for folding. This mechanism includes reducing new protein synthesis, increasing the ER’s protein-folding capacity, and promoting the degradation of misfolded proteins. The UPR plays a critical role in maintaining the balance of ER function. When stress is alleviated, the UPR signaling pathway is negatively regulated to restore normal ER function. However, prolonged or excessive ER stress can cause the UPR to shift from an adaptive response to an apoptotic signal. Extended UPR activation leads to the upregulation of pro-apoptotic factors, such as CHOP, triggering cell apoptosis (Huang et al., 2019). Furthermore, excessive activation of IRE1 can promote cell death through the c-Jun N-terminal kinase (JNK) signaling pathway (Brozzi et al., 2014; Zhang et al., 2024)

Regulation of ER stress in pancreatic islet cells

Calcium ion homeostasis is critical for the proper folding of proteins and the activity of enzymes within the ER. When calcium ions are imbalanced, whether through leakage or excessive accumulation, the ER’s function is disrupted, leading to protein misfolding (As illustrated in Fig. 2). The excessive activation of the UPR in response to ER stress helps restore calcium homeostasis by regulating the expression of calcium channels and pumps in the ER membrane (Engin et al., 2013). For instance, UPR activation can upregulate the expression of calcium-binding proteins and calcium transporters, facilitating the balance of calcium ions between the ER and the cytoplasm. Key UPR sensors like IRE1, PERK, and ATF6 play an important role in this process by activating downstream signaling pathways that regulate calcium regulation and homeostasis (Mori, 2022; Ramos-Castaneda et al., 2005). This regulation helps maintain an optimal calcium concentration within the ER, ensuring correct protein folding and processing (Read & Schroder, 2021). In pancreatic β-cells, calcium homeostasis is especially crucial because calcium ions are involved not only in insulin secretion but also in the ER’s protein-folding capacity and protein synthesis. Disruption of calcium balance exacerbates ER stress, impairs insulin synthesis and secretion, and contributes to β-cell dysfunction, which is closely associated with the development of insulin resistance (Emfinger et al., 2022; Sabatini, Speckmann & Lynn, 2019).

Figure 2 Activation of endoplasmic reticulum stress and UPR signaling pathways.

Disruption of calcium ion balance, ROS, accumulation of misfolded proteins, and increased protein synthesis load in the endoplasmic reticulum lead to the induction of ER stress, which subsequently activates the UPR through the IRE1, PERK, and ATF6 pathways. In the IRE1 pathway, IRE1 activates XBP1 mRNA, promoting its splicing into the active XBP1s form, which then enters the nucleus to regulate the expression of genes involved in molecular chaperones, ERAD factors, and lipid metabolism, thereby enhancing the ER’s protein folding and degradation capacity. In the PERK pathway, activation of PERK leads to the phosphorylation of eIF2α, which inhibits protein synthesis and reduces the load on the ER. Additionally, phosphorylated promotes the activation of ATF4, further enhancing the expression of genes related to antioxidant defense and stress resistance. In the ATF6 pathway, ATF6 is transported to the Golgi apparatus, where it is cleaved into its active form. This fragment then enters the nucleus and promotes the expression of genes related to ER protein folding, the ERAD system, and stress-responsive proteins.

Reactive oxygen species (ROS) are primarily generated by the mitochondrial electron transport chain and NADPH oxidases. In metabolically active cells, an increase in energy consumption often results in ROS accumulation, leading to oxidative stress (Zorov, Juhaszova & Sollott, 2014). This disrupts the redox balance within the ER, affecting the formation of disulfide bonds in proteins and interfering with correct protein folding (Wang et al., 2025). In the ER, ROS, particularly hydrogen peroxide (H2O2), are produced as byproducts during the formation of disulfide bonds in nascent proteins by protein disulfide isomerases (PDI) and other enzymes involved in protein folding. H2O2 is a relatively stable ROS that can accumulate within the ER lumen (Cheng et al., 2023). UPR counteracts oxidative stress by enhancing the expression of antioxidant genes, such as glutathione synthetase and superoxide dismutase, thereby boosting the cell’s antioxidant capacity. Additionally, UPR promotes NADPH production, providing the reducing power to counteract oxidative damage. These regulatory mechanisms help restore the ER’s redox balance, reduce ROS accumulation, and protect the protein-folding process (Ong & Logue, 2023). In β-cells, when there is an increased demand for insulin production, the overproduction of disulfide bonds to stabilize the insulin precursor molecules leads to a significant buildup of H2O2 (Vidrio-Huerta, Plotz & Lortz, 2024). This elevated H2O2 level can overwhelm the cell’s antioxidant defenses, contributing to oxidative stress and further disrupting ER function (Zito, 2015). Excessive ROS can damage the protein-folding machinery, particularly in the context of diabetes or β-cell dysfunction. Oxidative stress not only leads to the accumulation of misfolded proteins in the ER but also upregulates pro-apoptotic factors within the UPR, enhancing β-cell apoptosis and accelerating β-cell failure (Lenzen, 2017; Roohi et al., 2023).

In pancreatic β-cells, the activation of ER stress is closely linked to insulin synthesis and secretion. Chronic or excessive ER stress leads to the accumulation of misfolded proteins, causing the ER to lose its protein-folding capabilities, which results in impaired insulin synthesis and secretion. UPR alleviates this stress by reducing protein translation rates, thus decreasing the load of nascent proteins entering the ER (Hetz, 2012). For example, UPR regulates the activity of translation initiation factors in β-cells, temporarily inhibiting protein synthesis. Simultaneously, UPR upregulates the expression of molecular chaperones and folding enzymes, enhancing the ER’s protein-folding capacity. Moreover, UPR activates the ER-associated degradation (ERAD) pathway, accelerating the recognition and degradation of misfolded proteins to prevent their accumulation within the ER (Maillo et al., 2017). Cells also enhance the expression of molecular chaperones, such as BiP/GRP78 and protein disulfide isomerase, to accelerate correct protein folding in β-cells, thereby restoring ER function in response to ER stress during the progression of diabetes

The three signaling pathways of the UPR in islet cells

The UPR is mediated by three major signaling pathways, each initiated by distinct receptors located on the ER membrane: IRE1, PERK, and ATF6. These receptors are activated in response to the accumulation of unfolded proteins, triggering their respective downstream signaling cascades.

IRE1 is a transmembrane protein with both kinase and ribonuclease (RNase) activities. Upon activation during ER stress, the kinase domain of IRE1 is triggered, leading to RNase activation and the unconventional splicing of XBP1 mRNA. The spliced form, XBP1s (the active transcription factor), translocates to the nucleus and regulates the expression of ER-related genes, including molecular chaperones, ERAD factors, and genes involved in lipid metabolism, thereby enhancing the protein-folding and degradation capabilities of the ER. In β-cells, the absence of XBP1s results in defects in proinsulin processing, impaired insulin secretion, and a loss of β-cell adaptive proliferation, leading to reduced glucose tolerance and hyperglycemia (Lee et al., 2011; Lee et al., 2022). Conversely, sustained expression of XBP1s in insulin-secreting cells can impair insulin secretion and promote β-cell apoptosis (Allagnat et al., 2010).

PERK is another key sensor in the ER stress response. Its activation leads to the phosphorylation of eIF2α, significantly inhibiting global protein synthesis. This mechanism effectively diminishes the influx of nascent proteins into the ER, thus alleviating the folding burden on the organelle (Merrick & Pavitt, 2018). In addition to alleviating the ER load, eIF2α phosphorylation promotes the selective translation of ATF4. In the nucleus, ATF4 regulates the expression of genes involved in amino acid metabolism, oxidative stress response, and apoptosis, thereby helping cells adjust their metabolic pathways and enhance antioxidant and stress-resistance capabilities in response to stress (Gao et al., 2019; Lee & Ozcan, 2014). In β-cells, PERK deficiency results in ER stress, β-cell loss, and severe hyperglycemia (Delepine et al., 2000; Zhang et al., 2006). Moreover, ATF4 deficiency exacerbates hyperglycemic symptoms in Akita mice, indicating the critical role of the PERK-ATF4 pathway in β-cell survival and function (Kitakaze et al., 2021). Deletion of downstream PERK factors, such as CHOP, TRIB3, and 4E-BP1, can mitigate ER stress and prevent β-cell loss (Oyadomari et al., 2002; Song et al., 2008; Yong et al., 2021b).

ATF6 is translocated from the ER to the Golgi apparatus during ER stress, where it is cleaved by proteases to release an N-terminal fragment with transcriptional activity. This fragment enters the nucleus and regulates the expression of genes related to ER protein folding, the ERAD system, and stress-responsive proteins (Nozaki et al., 2004; Sharma et al., 2015). By promoting the synthesis of molecular chaperones, folding enzymes, and ERAD factors, ATF6 effectively enhances the ER’s capacity to process misfolded proteins, thereby improving protein-folding efficiency. This regulatory mechanism helps maintain the balance of protein folding under stress conditions and prevents the excessive accumulation of misfolded proteins, thus preserving cellular function. In β-cells, ATF6 supports insulin folding and processing by promoting the expression of molecular chaperones and folding enzymes, thereby maintaining insulin synthesis and secretion (Sharma et al., 2015). Additionally, early insulin demand and UPR activation mediated by ER stress have been reported to promote β-cell proliferation (Sharma et al., 2015).

ER Stress Regulation in Pancreatic Islet Cells

ER stress impairs β-cell insulin synthesis and secretion

In the early stages of T2DM, peripheral insulin resistance increases the demand for insulin, which in turn induces ER stress in pancreatic β-cells. As previously discussed, the UPR is activated in β-cells, enhancing protein folding and the degradation of misfolded proteins. This process involves the IRE1, ATF6, and PERK signaling pathways (Delepine et al., 2000; Szabat et al., 2016; Wang & Kaufman, 2016; Xiong et al., 2020), coordinating cellular responses to restore homeostasis (Eizirik, Cardozo & Cnop, 2008; Xiong et al., 2020). These pathways synergistically upregulate the expression of molecular chaperones, such as GRP78 (BiP), to assist in protein folding and activate the ERAD pathway to eliminate misfolded proteins (Eizirik, Cardozo & Cnop, 2008). Notably, both XBP1 and ATF6 pathways are simultaneously activated in response to acute stress in pancreatic islet cells, and may act in concert during transcriptional regulation (Sharma, Darko & Alonso, 2020). Moreover, the UPR can promote autophagy, a cellular process that removes damaged proteins, further alleviating the burden of ER stress (Eizirik, Cardozo & Cnop, 2008). However, when ER stress becomes chronic and the UPR is suppressed, autophagy may be insufficient, leading to β-cell dysfunction and apoptosis (Eizirik, Cardozo & Cnop, 2008; Omar-Hmeadi & Idevall-Hagren, 2021). Despite these adaptive mechanisms, prolonged ER stress can overwhelm the UPR, resulting in limited insulin synthesis and secretion (Omar-Hmeadi & Idevall-Hagren, 2021; Sharma et al., 2015).

Studies have shown that acute reductions in insulin synthesis can alleviate ER stress in β-cells, promoting cell proliferation and functionality (Omar-Hmeadi & Idevall-Hagren, 2021; Xiong et al., 2020). This suggests that regulating insulin biosynthesis and enhancing β-cell stress tolerance are critical for maintaining cellular health. Temporarily decreasing insulin production may help prevent the accumulation of unfolded proteins, allowing the UPR to restore cellular function and ultimately maintain β-cell survival. Some studies further suggest that transient reductions in insulin production can induce a “re-folding” response within the ER, promoting better management of unfolded proteins and enhancing β-cell survival (Xiong et al., 2020). Additionally, the degree of ER stress in β-cells may influence the expression of key transcription factors, such as NKX6.1 and MAFA, which may not only affect β-cell identity but also significantly influence endogenous insulin synthesis. During this process, the ratio of preinsulin/insulin within the cell is altered, indicating impaired β-cell synthetic capacity (Brusco et al., 2023). This could be linked to the disruption of redox homeostasis within the β-cell ER, where excessive oxidation in the ER lumen compromises preinsulin folding and disulfide bond formation, thus impairing the effective export of preinsulin from the ER to the Golgi apparatus (Rohli et al., 2022).

It is well-established that Ca2+ plays a crucial role in insulin secretion, and ER stress significantly decreases ER Ca2+ levels. In the early stages of ER stress, Ca2+ influx through store-operated calcium entry (SOCE) channels leads to fluctuations in intracellular Ca2+ concentrations, thereby enhancing insulin secretion (Zhang et al., 2020). Moreover, the deletion of IRE1α and XBP1 in mouse β-cells diminishes the number of insulin granules and impairs glucose-stimulated insulin secretion (Lee et al., 2011). Prolonged stress conditions disrupt the metabolic and electrophysiological functions of β-cells, leading to impaired glucose-stimulated insulin secretion (Fang et al., 2019; Farack et al., 2019). This impairment is partly attributed to alterations in intracellular calcium dynamics, which are essential for the exocytosis of insulin granules (Ravier et al., 2011; Tong et al., 2016). Chronic ER stress further disrupts proinsulin folding efficiency. Increased insulin demand accelerates proinsulin synthesis, which may surpass the ER’s folding capacity. Misfolded proinsulin aggregates, linked by disulfide bonds, accumulate in the ER, exacerbating stress and further impairing insulin secretion. For instance, in the db/db mouse model, prolonged ER stress leads to decreased insulin content due to impaired preinsulin folding and increased degradation (Arunagiri et al., 2019; Haataja et al., 2021; Liu et al., 2018; Schuit, Kiekens & Pipeleers, 1991).

Mechanisms by which ER stress leads to β-cell apoptosis

Chronic ER stress not only impairs insulin synthesis but also activates apoptotic pathways in β-cells (as shown in Fig. 3). Prolonged activation of the UPR can shift from a protective mechanism to one that promotes cell death. A key factor in this process is the transcription factor CHOP, which is upregulated during prolonged ER stress (Song et al., 2008). The increased expression of CHOP regulates apoptosis-related genes, including downregulating the anti-apoptotic protein Bcl-2 and upregulating pro-apoptotic factors (Harding et al., 2009). In the context of type 2 diabetes, the db/db mouse model exhibits CHOP-dependent β-cell death, where deletion of the Chop gene lowers ER stress and oxidative stress, protecting β-cell mass and function (Song et al., 2008).

Figure 3 Impact of endoplasmic reticulum stress on insulin secretion.

Excessive insulin secretion by pancreatic islet cells increases ER stress, leading to the accumulation of misfolded proteins and triggering the UPR. The UPR degrades misfolded proteins to alleviate ER stress. However, when the ER’s capacity for regulation is exceeded, CHOP is activated, promoting ROS generation and disrupting calcium ion homeostasis, which ultimately leads to β-cell apoptosis and reduced insulin synthesis.

Oxidative stress, another mechanism induced by ER stress, also contributes to β-cell apoptosis. The accumulation of misfolded proteins in the ER generates ROS, leading to cellular damage and cell death (Jang et al., 2019). β-cells are particularly susceptible to oxidative stress due to their relatively low levels of antioxidant enzymes (Robertson, 2004). This oxidative stress exacerbates ER stress, creating a harmful feedback loop that accelerates β-cell death. Studies suggest that ubiquitination may also play a role in regulating oxidative stress in β-cells. For instance, the accumulation of misfolded islet amyloid polypeptide (IAPP) leads to the loss of ubiquitin carboxyl-terminal hydrolase L1 (UCH-L1), which disrupts the ER-associated degradation system, increases polyubiquitinated protein accumulation, and triggers both ER stress and β-cell apoptosis (Costes et al., 2011).

Additionally, dysregulation of ER calcium homeostasis plays a critical role in β-cell apoptosis. Proper calcium levels within the ER are essential for protein folding and signal transduction. ER stress disrupts calcium balance, triggering calcium-dependent apoptotic pathways (Gelebart, Opas & Michalak, 2005; Michalak et al., 1999; Suzuki et al., 1991). For example, mutations in the sarco/endoplasmic reticulum calcium ATPase 2 (SERCA2) impair the function of calcium pumps in the ER, making β-cells more prone to apoptosis induced by ER stress (Johnson et al., 2014). Therefore, under hyperglycemic conditions, the calcium buffering and storage function of the ER may be compromised, leading to increased ER stress and β-cell death (Suzuki et al., 1991). Furthermore, the mechanism by which ATP enters the ER is closely linked to cytosolic calcium concentrations, further influencing β-cell responses to ER stress (Yong et al., 2019).

Moreover, advanced glycation end products (AGEs), which accumulate under prolonged hyperglycemic conditions, have been proposed as a potential mechanism for ER stress-mediated β-cell death (Liu et al., 2017). AGEs may trigger the initiation of the ER stress response by interacting with β-cell proteins, leading to cellular damage. Although the role of AGEs in ER stress-induced β-cell death is relatively minor compared to other mechanisms, the absence of fructosamine-3-kinase (FN3K), an enzyme critical for protein deglycosylation, does not exacerbate β-cell glucotoxicity (Pascal et al., 2010). However, the administration of AGEs can exacerbate β-cell ER stress, further contributing to cellular dysfunction and death (Piperi et al., 2012).

ER stress in alpha cells

Studies have shown that in patients with T2DM, the ER volume density in both α-cells and β-cells in the pancreas increases more than threefold, indicating that these cells are under functional overload and have activated a compensatory UPR (Marroqui et al., 2015). A similar phenomenon was observed in isolated islets exposed to palmitic acid, a saturated fatty acid associated with T2DM-related metabolic stress. Both α-cells and β-cells showed significant increases in ER volume density (Marroqui et al., 2015). Studies show that in triglyceride-induced α-cell stress, the number of differentially expressed genes in α-cells is much higher than in β-cells and δ-cells, suggesting that α-cells may adapt to the stress environment by altering their metabolism and structure (Maestas et al., 2024). Interestingly, while both α-cells and β-cells experienced ER stress under these conditions, only β-cells exhibited more than a fourfold increase in apoptosis, while α-cells remained completely viable (Marroqui et al., 2015). This suggests that although both cell types undergo ER stress in diabetes, β-cells are more vulnerable to ER stress-induced death.

Previous studies have indicated that α-cells have higher levels of the chaperone protein BiP (which assists in protein folding in the ER) and the anti-apoptotic protein BCL2L1 (Marroqui et al., 2015). BCL2L1 plays a critical role in protecting α-cells from apoptosis; inhibition of its expression makes α-cells more susceptible to palmitic acid-induced apoptosis, similar to the response of β-cells to this stressor (Marroqui et al., 2015). These findings suggest that α-cells possess stronger ER stress protection mechanisms compared to β-cells. Islet α-cells exhibit significant transcriptional changes in response to ER stress, including the upregulation of various ER stress-related genes such as ARF4, CREB3, and COG6 in response to brefeldin A-induced stress. Additionally, α-cells show higher gene expression changes, particularly in glycolysis, and respond to ER damage by regulating ER stress-related genes such as CRELD2, JUN, and COX14, suggesting that α-cells may protect themselves from stress-induced damage through these mechanisms (Maestas et al., 2024).

Single-cell RNA sequencing studies further confirm these observations when human islet cells are exposed to IFNα.α-like cells derived from induced pluripotent stem cells (iPSCs) exhibited significantly higher levels of BiP and BCL2L1 expression compared to β-like cells (Szymczak et al., 2022). Additionally, another scRNA-seq study observed an increase in BiP expression in α-cells in both normal blood glucose individuals and T1DM patients (Chen et al., 2022).

ER stress in type 1 and type 2 diabetes: a comparative overview

Extensive research over the past 15 years has consistently shown that ER stress plays a pivotal role in the dysfunction and failure of pancreatic β-cells in both type 1 diabetes mellitus (T1DM) and T2DM. While ER stress is a common feature of both conditions, the underlying mechanisms and signaling pathways that lead to β-cell dysfunction and eventual cell death differ markedly between T1DM and T2DM.

In T1DM, autoimmune destruction of β-cells is the hallmark of the disease (Katsarou et al., 2017). Pro-inflammatory cytokines released by infiltrating immune cells trigger ER stress within β-cells, including cytokines such as IFNα,IFNγ, IL-1β, and IL-17, which exhibit differential expression during various stages of disease progression (Eizirik, Pasquali & Cnop, 2020; Eizirik et al., 2012; Ortis et al., 2010). Additionally, amyloid deposits have been observed in the islets of some patients with recent-onset T1DM, and these deposits may contribute to ER stress (Beery et al., 2019; Westermark et al., 2017). The stress is primarily mediated by the hyperactivation of the IRE1α pathway, which significantly impacts β-cell viability (Ghosh et al., 2014). Upon activation, IRE1α undergoes homodimerization and trans-autophosphorylation, enhancing its endoribonuclease (RNase) activity. This leads to the splicing of XBP1 mRNA, producing the active XBP1s transcription factor, which upregulates genes involved in ER expansion and chaperone production (Ghosh et al., 2014; Li et al., 2022). However, excessive IRE1α activity, particularly during severe ER stress, also triggers regulated IRE1-dependent decay (RIDD), which degrades ER-localized mRNAs and activates the JNK pathway, ultimately promoting apoptosis (Cnop et al., 2017; Sano & Reed, 2013). Moreover, studies have shown that low levels of IL-1β and mild ER stress enhance β-cell inflammation via the IRE1α/XBP1 pathway, further aggravating β-cell damage (Miani et al., 2013).

In NOD mice, the translocation of ABL tyrosine kinase to the ER membrane enhances IRE1’s RNase activity, promoting β-cell apoptosis (Morita et al., 2017). In response to this excessive activation, β-cells have intrinsic protective mechanisms, including the N-MYC interacting factor and Ubiquitin D, which feedback-regulate IRE1-induced JNK activation to partially prevent excessive β-cell death (Brozzi et al., 2016; Brozzi et al., 2014). However, these protective mechanisms are insufficient to counter prolonged or persistent immune-mediated attack. Interestingly, by modulating IRE1 hyperactivation, β-cell survival and function can be improved. For example, KIRA6 and Imatinib (an FDA-approved tyrosine kinase inhibitor) have been shown to inhibit IRE1’s RNase activity and restore glucose homeostasis in NOD mice (Ghosh et al., 2014; Morita et al., 2017).

In contrast, T2DM is characterized by chronic metabolic stressors, including hyperglycemia and elevated free fatty acid (FFA) levels, both of which contribute to ER stress in β-cells. High glucose concentrations increase insulin demand, exerting considerable pressure on the ER’s protein-folding capacity. This condition induces mild ER stress, primarily activating the IRE1 and activating transcription factor 6 (ATF6) pathways (Elouil et al., 2007; Lipson et al., 2006). However, saturated FFAs, such as palmitate, trigger a more pronounced activation of the protein kinase RNA-like ER kinase (PERK) pathway, which leads to phosphorylation of eukaryotic initiation factor 2 alpha (eIF2α) and subsequently decreases global protein synthesis to alleviate the ER load. Prolonged activation of PERK, however, results in the increased expression of the pro-apoptotic transcription factor CHOP, thereby promoting β-cell apoptosis. Evidence from T2DM patients shows elevated markers of ER stress in β-cells, such as increased levels of p58IPK, ATF3, CHOP, and BiP, indicative of PERK pathway activation (Huang et al., 2007; Laybutt et al., 2007). Interestingly, levels of ATF6 and XBP1s proteins are found to be decreased in T2DM islets, suggesting that the adaptive response is diminished, and the ability of β-cells to cope with ER stress is compromised (Engin et al., 2014).

Moreover, the role of local inflammation in exacerbating β-cell ER stress in T2DM cannot be overlooked. Chronic low-grade inflammation, a hallmark of T2DM, involves the infiltration of immune cells into the islets and the secretion of pro-inflammatory cytokines, including IL-1β, IL-6, and TNF-α. These cytokines further contribute to β-cell dysfunction by amplifying ER stress through various mechanisms. For instance, IL-1β has been shown to downregulate ER Ca2+ storage, thereby exacerbating ER stress in β-cells. The combined effects of metabolic stress and inflammation create a “toxic” microenvironment that impairs the ability of β-cells to respond to glucose, ultimately leading to β-cell death (O’Neill et al., 2013; Westwell-Roper, Ehses & Verchere, 2014). Interestingly, studies have highlighted the dual role of inflammation in T2DM: while certain pro-inflammatory cytokines, like IL-1β, contribute to β-cell dysfunction, others, such as IL-10 and IL-22, appear to exert protective effects. These cytokines can alleviate ER stress in β-cells, reducing apoptosis and improving β-cell function. The balance between these opposing cytokines may be a critical determinant of the severity of β-cell dysfunction in T2DM, illustrating the complex interplay between metabolic stress, inflammation, and ER stress pathways (Hasnain et al., 2014; Wang et al., 2014).

Thus, while ER stress is a shared mechanism underlying β-cell failure in both T1DM and T2DM, the distinct etiologies and immune responses in these two conditions lead to differential patterns of β-cell stress, dysfunction, and death. Understanding these differences is critical for developing tailored therapeutic strategies aimed at alleviating ER stress and preserving β-cell function in both forms of diabetes.

ER stress in endothelial cells and its impact on islet cell function

In recent research, the relationship between ER stress in endothelial cells and the function of islet cells in the pancreas has gained increasing attention (Craig-Schapiro et al., 2025). Endothelial cells not only form the microvascular endothelial barrier that supports normal pancreatic function but also interact with islet cells to influence insulin secretion and the overall metabolic state of the pancreas. Studies have shown that endothelial cells play a crucial role in the interplay between oxidative stress and ER stress within the pancreas, which significantly affects islet cell health and function (Jonsson et al., 2020). The molecular interactions between islets and islet endothelial cells contribute to the induction, differentiation, and maintenance of islet function, particularly in insulin gene expression and glucose-stimulated insulin secretion in β-cells (Obata et al., 2019). Endothelial cells regulate blood flow, angiogenesis, and act as an immune barrier between the blood and pancreatic tissue. Vascular endothelial growth factor (VEGF) is highly expressed in β-cells, promoting endothelial cell proliferation and supporting islet vascularization (Li et al., 2017; Staels et al., 2019). Inhibition of VEGF signaling can lead to the regression of islet vasculature, while VEGF overexpression can improve islet quality.

Oxidative stress induced by hyperglycemia and hyperlipidemia generates ROS in endothelial cells. These ROS disrupt the redox balance within the ER, promoting the accumulation of misfolded or unfolded proteins, which in turn triggers the ER stress response. Together with increased oxidative stress, this mediates endothelial dysfunction, which is an initiating event for diabetes-mediated complications (Ogita & Liao, 2004). It has been shown that elevated oxidative stress, along with increased concentrations of the vasodilator NO⋅, triggers its bioactive form, cyclic 3′, 5′-guanosine monophosphate (cGMP), which regulates endothelial dysfunction in STZ-induced diabetes (Bojunga et al., 2004). Moreover, it has been reported that increased NO⋅ can promote apoptosis, intervene in endothelial injury (Rydgren & Sandler, 2002), and induce β-cell dysfunction in STZ-induced diabetic models, with its effects triggered through the activation of endoplasmic reticulum (ER) stress. Furthermore, ER stress and the unfolded protein response (UPR) are induced by the effects of reactive oxygen species (ROS) and NO⋅, linking the upregulation of inflammation and stress signaling networks (Suganya et al., 2018). The ER stress response includes the activation of the UPR, which aims to restore normal protein folding. However, this process can also initiate cellular inflammation and other adverse physiological changes. In pancreatic endothelial cells, ER stress and oxidative stress often mutually enhance each other, with the NOX4 enzyme playing a pivotal role in generating H2O2 to further exacerbate oxidative stress (Wang & Zhang, 2021).

At the molecular level, ER stress activates several signaling pathways that alter endothelial cell function. First, ER stress induces the leakage of Ca2+ from the ER into the cytoplasm. This not only changes the interaction between the endothelial cell ER and mitochondria but also activates downstream signaling pathways such as calcium/calmodulin-dependent protein kinase II (CaMKII) and protein kinase C (PKC). The activation of these pathways affects endothelial cell physiology, particularly its support of islet cells (Fu et al., 2011). Secondly, ER stress generates ROS, which activate inflammation-related signaling pathways, such as the NF-κB pathway. The activation of NF-κB promotes the expression of pro-inflammatory cytokines, causing an inflammatory response in endothelial cells, which disrupts the islet cell microenvironment and negatively impacts β-cell function (Maamoun et al., 2019; Sipkens et al., 2013).

Furthermore, the interaction between ER stress and oxidative stress leads to the overactivation of the NOX4 enzyme, which generates additional ROS. These ROS not only directly damage endothelial cells but also amplify the ER stress response, creating a vicious cycle that further exacerbates endothelial dysfunction. NOX4, by producing H2O2, increases oxidative stress within the cell. This not only disrupts the structure and function of endothelial cells but also alters the redox state of the islet cell microenvironment, thereby suppressing insulin secretion by islet cells (Wang & Zhang, 2021).

Chronic ER stress and oxidative stress can lead to the degeneration of endothelial cell function, including increased vascular permeability, endothelial cell apoptosis, and dysfunction. These changes result in structural and functional abnormalities in the pancreatic microvasculature. Such endothelial dysfunction not only affects the blood supply to the pancreas but also alters the metabolic environment of islet cells, indirectly suppressing their insulin secretion capacity and worsening the progression of diabetes.

Conclusions

This review highlights the critical role of ER stress in T2DM, particularly its impact on the function and survival of islet, especially pancreatic β-cells. β-cells, which are responsible for synthesizing and secreting large amounts of insulin to maintain glucose homeostasis, are especially sensitive to ER stress. This stress is primarily caused by the accumulation of unfolded or misfolded proteins within the cell. In the early stages of T2DM, increased insulin demand due to insulin resistance exacerbates ER stress in β-cells. Although cells initiate the UPR to cope with this stress, prolonged or excessive ER stress exceeds the cell’s adaptive capacity, leading to β-cell dysfunction, apoptosis, and ultimately insufficient insulin secretion, which contributes to the progression of T2DM.

In summary, a deeper understanding of how ER stress affects islet cells, along with the development of targeted therapeutic strategies, may provide new avenues for treating T2DM. One key area for future research is the identification and development of specific molecular regulators that can modulate the UPR, particularly in a way that maintains β-cell function without triggering apoptosis. This could involve exploring the roles of specific signaling pathways like IRE1, PERK, and ATF6, and how their modulation can prevent the transition from adaptive UPR responses to apoptosis. Additionally, understanding the differences in ER stress responses between β-cells, α-cells, and other islet cells could reveal new therapeutic targets to improve insulin production and secretion in T2DM patients. The interactions of ER stress with other cellular pathways involved in metabolism and inflammation should also be studied in depth, as these pathways may offer synergistic or antagonistic effects that impact islet cell function and overall disease progression.

Furthermore, reducing ER stress and enhancing β-cell stress resistance may offer improved treatment outcomes and better quality of life for diabetes patients. Emerging therapeutic approaches, such as RNA therapies and gene editing tools, present exciting new possibilities. RNA-based treatments, like siRNA and mRNA therapies, could directly target stress-related pathways, offering precise modulation of β-cell function. Gene editing technologies like CRISPR-Cas9 could be used to correct stress-related genetic mutations or enhance β-cell resilience, potentially reversing dysfunction in T2DM patients. Future research should explore the clinical applications and long-term effects of these strategies, particularly regarding their ability to prevent or reverse β-cell dysfunction in patients with chronic T2DM. Moreover, the potential of natural compounds, which may act as ER stress modulators, should also be investigated, as these may provide safer and more cost-effective treatments. Identifying and validating specific biomarkers for in vivo monitoring of ER stress could be a critical step in developing precise, individualized therapies to manage β-cell health.

However, there are several key challenges that remain in understanding the full scope of ER stress in diabetes. One major challenge is determining how to precisely control the balance between beneficial UPR activation and harmful chronic stress responses in β-cells. It is also crucial to develop strategies that can address the systemic effects of ER stress, including the influence of metabolic and inflammatory stressors that compound β-cell damage in T2DM. Additionally, exploring the role of ER stress in non-β islet cells—such as α-cells, δ-cells, and endothelial cells—remains underexplored and could provide valuable insights into their contribution to glucose homeostasis and pancreatic function in diabetes. Future research should aim to bridge the gap between experimental models and clinical application, with a focus on patient-specific factors that might influence the effectiveness of potential therapies. Understanding these complexities will be essential to designing novel therapeutic interventions that not only address the metabolic symptoms of T2DM but also target the root causes of β-cell dysfunction. By combining advances in molecular biology, pharmacology, and clinical research, we can better address the challenge of ER stress in diabetes and create more effective strategies for its management.

Additional Information and Declarations

Competing Interests

Author Contributions

Data Availability

The authors declare there are no competing interests.

Zhaxicao He conceived and designed the experiments, performed the experiments, analyzed the data, prepared figures and/or tables, authored or reviewed drafts of the article, and approved the final draft.

Qian Liu conceived and designed the experiments, performed the experiments, prepared figures and/or tables, authored or reviewed drafts of the article, and approved the final draft.

Yan Wang performed the experiments, prepared figures and/or tables, authored or reviewed drafts of the article, and approved the final draft.

Bing Zhao performed the experiments, prepared figures and/or tables, authored or reviewed drafts of the article, and approved the final draft.

Lumei Zhang performed the experiments, prepared figures and/or tables, authored or reviewed drafts of the article, and approved the final draft.

Xia Yang performed the experiments, prepared figures and/or tables, authored or reviewed drafts of the article, and approved the final draft.

Zhigang Wang performed the experiments, analyzed the data, prepared figures and/or tables, authored or reviewed drafts of the article, and approved the final draft.

The following information was supplied regarding data availability:

This is a literature review.

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
