# Peer review of "The role of endoplasmic reticulum stress in type 2 diabetes mellitus mechanisms and impact on islet function"

_PeerJ, doi:10.7717/peerj.19192_

## Round 0.1 · original submission · Minor Revisions

As you will see, both reviewers liked your review but have offered some suggestions which I'd like you to address. Please explain your responses to each point in a cover letter and address each in your revised paper.

Reviewer 1 ·

Basic reporting

This review provides a good overview of how ER stress contributes to the disease mechanisms of diabetes. This is a complex topic and the authors describe in detail what is understood across different types of islet cells, with reference to many experimental studies, which will provide a good overview to those from a diabetes background who are not familiar with ER stress mechanisms. I haven’t come across a review that compare ER stress mechanisms across different islet cell types, so I believe that this review fills a gap in the literature.

In paragraph 2 of the introduction the ER is introduced. I don’t think enough description is given here about the function and complexity of the ER to adequately introduce the topic. This should be expanded upon to emphasise that the ER (i) is a specialised folding environment that contrasts to the cytosol, with a redox balance and many enzymes that assist the folding of proteins, (ii) that it is the major site of disulfide bond formation in the cell and (iii) it is the major entry point to the secretory pathway, where almost all secretory proteins fold.

I also think that adding in details about the biosynthesis and structure of insulin, would strengthen the introduction. For example, the disulfide bonds that form as insulin folds in the ER are important details that can contribute to ER stress.

Experimental design

The review is logically structured. I have made a few suggestions in the general comments section.

It would be nice to have more citations from research groups that elucidated mechanism of the UPR. Such as Peter Walter, David Ron, and Kazutoshi Mori.

Validity of the findings

A few lines to explain the aims and purpose of the review at the end of the introduction would be useful.

The conclusion is very short. I would like to see the authors describe where they see the field going in the future and any major questions that still need to be addressed, in order to better understand the role of ER stress in diabetes.

Additional comments

Lines 129-130: The description here implies that all proteins fold in the ER. It should be pointed out here that all proteins are synthesised on cytosolic ribosomes, and a large subset (but not all) are targeted to the ER for folding.

Line 143: IREI, PERK and ATF6 should be mentioned and briefly introduced at this earlier point.

Somewhere in section 3. there needs to be a paragraph on production of ROS in the ER, the major one being H202. H202 is a by product of disulfide formation and so when insulin is being over produced this will be a major contributor to ER stress. I don’t think this is mentioned throughout the review, and it is very important reason why excess insulin production leads to ER stress.

The term “reduces” is used to describe a decrease on multiple occasions. Examples include line 45 and 205. This should be avoided in publications about the ER as it gets confused with reduction/oxidation terminology.

Lines 140-141 should read “the excessive activation of UPR in response to ER stress.

·

Basic reporting

• Language and Clarity: The manuscript is written in clear and professional English, suitable for an international audience. However, there are some minor grammatical errors and instances of awkward phrasing, such as repetitive sentence structures in the introduction. Polishing these areas would enhance readability.
• Introduction and Background: The introduction provides a comprehensive overview of Type 2 Diabetes Mellitus (T2DM) and the critical role of endoplasmic reticulum (ER) stress in β-cell dysfunction. The background information is detailed and well-cited. However, the manuscript would benefit from a more explicit statement of the knowledge gap that this review aims to address. Additionally, a clearer explanation of why this review is timely and how it contributes a novel perspective would strengthen the introduction.
• References: References are relevant and up-to-date, citing seminal studies as well as recent advances. Some references, however, are repeated unnecessarily within sections (e.g., citations for CHOP and UPR pathways). Streamlining these references can improve flow.
• Structure and Formatting: The manuscript adheres to standard academic structure, with logical subdivisions and clear section headings. Figures and tables are appropriate and enhance the narrative, though their captions could be more descriptive. For example, Figure 2 could benefit from a more detailed explanation of the interplay between UPR pathways.

Experimental design

• Literature Survey: The authors demonstrate a thorough review of the literature, utilizing a broad range of studies to construct their arguments. However, the inclusion criteria for selecting references should be explicitly mentioned. For instance, were studies prioritized based on recency, relevance, or methodology?
• Coverage: While the review is comprehensive in addressing ER stress mechanisms in β-cell dysfunction, it underrepresents certain emerging therapeutic strategies, such as RNA-based interventions and novel small molecules targeting ER stress pathways. Including these could make the manuscript more current and impactful.
• Organization: The review is well-organized, but some sections, such as "Mechanisms of ER Stress" and "Therapeutic Strategies," are disproportionately detailed compared to others like "ER Stress in Alpha Cells." Balancing the depth across sections would improve coherence.

Validity of the findings

• Evidence and Argumentation: The manuscript builds a solid case for the role of ER stress in T2DM by synthesizing findings from multiple studies. The mechanisms of ER stress, including UPR pathways, CHOP-mediated apoptosis, and the interplay with oxidative stress, are well-articulated. However, some conclusions, such as the protective role of UPR activation, are presented as broadly applicable without acknowledging the variability across experimental models and clinical scenarios. More nuanced discussion here would be beneficial.
• Unresolved Questions: The manuscript highlights gaps in understanding, such as the limited efficacy of current ER stress-targeting therapies. While this is commendable, it lacks concrete suggestions for future research directions. For instance, identifying specific biomarkers to monitor ER stress in vivo or exploring the crosstalk between ER stress and other metabolic pathways could be suggested.
• Novelty: The review provides a detailed summary of existing knowledge but does not significantly challenge prevailing perspectives or offer groundbreaking hypotheses. Emphasizing less-studied aspects, such as the role of ER stress in non-β islet cells, could enhance its originality.

Additional comments

Strengths:
1. Comprehensive coverage of ER stress mechanisms and their implications for β-cell function in T2DM.
2. Clear and logical structure with appropriate use of figures and tables.
3. Extensive and up-to-date referencing, including key foundational studies.

Areas for Improvement:
1. Introduction: Clearly articulate the novel contribution of this review and the knowledge gap being addressed.
2. Balance: Provide more balanced coverage across sections. For example, expand discussions on alpha cells and endothelial cells, as well as less-explored therapeutic approaches.
3. Therapeutic Strategies: Include emerging approaches, such as RNA-based therapeutics or gene-editing tools, and discuss their potential.
4. Figures: Improve figure captions to make them self-contained and more informative.
5. Language: Address minor grammatical issues and repetitive phrasing to enhance readability.

---

## Round 0.2 · accepted · Accept

Thank you for addressing the comments. I am happy to accept this now.

Reviewer 1 ·

Basic reporting

No comment

Experimental design

No comment

Validity of the findings

No comment

Additional comments

I have re-read the revised version of the article and I have found that:

The additions to the introduction significantly improve this section and provide a better background to the topic.

The additional citations throughout the article, give greater depth to the review, and make it more accessible.

The conclusion is much stronger with the additional sections added.

Overall the authors have listened to the opinions of the reviewers and revised the article appropriately. It is a significant improvement on the previous version and I recommend it for publication.